# Therapeutic Effects of Proanthocyanidins on Diabetic Erectile Dysfunction in Rats

**DOI:** 10.3390/ijms252011004

**Published:** 2024-10-13

**Authors:** Xiaoyan Zeng, Lanlan Li, Li Tong

**Affiliations:** 1Qinghai University, Xining 810000, China; zhcxbdy@163.com (X.Z.); hlanlan0169@163.com (L.L.); 2Qinghai Provincial Key Laboratory of Traditional Chinese Medicine Research for Glucolipid Metabolic Diseases, Xining 810000, China

**Keywords:** diabetic erectile dysfunction, proanthocyanidins, glucose and lipid metabolism, AKT1, CASP3, PI3K-Akt signaling pathway

## Abstract

The rising occurrence of erectile dysfunction related to diabetes mellitus (DMED) has led to the creation of new medications. Proanthocyanidins (PROs) is a potential agent for DMED. In this study, the DMED rat model was established using streptozotocin (STZ) and erectile function was assessed using apomorphine (APO) in rats. Following this, the rats were subjected to oral treatment with PRO. Then, we evaluated the influence of PROs on DMED rats. The findings suggest that PROs significantly enhance erectile function in DMED rats. PROs modulated glucose and lipid metabolism in DMED rats by decreasing blood glucose and lipid levels while increasing liver glycogen and serum insulin levels. Furthermore, PROs enhanced vascular endothelial function in DMED rats by augmenting nitric oxide (NO) levels and reducing the levels of endothelin-1 (ET-1) and lectin-like oxidized low-density lipoprotein receptor-1 (LOX-1). Additionally, PROs have been shown to elevate testosterone (T) levels, mitigate pathological testicular damage, and enhance sperm concentration and survival rates. Finally, the core targets were screened using network pharmacology, followed by validation through molecular docking, enzyme-linked immunoassay (ELISA), and real-time PCR methodologies. Our findings imply that PROs may treat DMED by elevating AKT1 levels while concurrently diminishing CASP3 levels, thereby effectively regulating the PI3K-Akt signaling pathway. Overall, these results support using PROs as a potential candidate for the treatment of DMED.

## 1. Introduction

Erectile dysfunction (ED) is a prevalent complication associated with diabetes mellitus (DM), defined as a man’s inability to achieve or maintain sufficient erectile rigidity during sexual intercourse [1]. ED resulting from diabetes mellitus is referred to as diabetic erectile dysfunction (DMED) [2]. Individuals with diabetes tend to develop ED at an earlier age compared to the general population [3]. Currently, phosphodiesterase type 5 inhibitors (PDE5i) represent the primary pharmacological intervention for the treatment of DMED [4]. However, diabetes can cause endothelial damage. Patients with DMED are significantly less sensitive to PDE5i inhibitors than non-diabetic patients and have greater side effects. Therefore, discovering a more effective therapy for DMED has become an urgent priority.

Surveys indicate that approximately one-third of patients diagnosed with DM utilize alternative medicine to regulate their blood glucose levels [5,6,7,8,9]. Herbal medicines are experiencing a significant rise in popularity globally for diabetes. The genus *Rubus*, belonging to the Rosaceae family, is extensively cultivated across Asia, Europe, and North America [10]. Several studies have demonstrated that raspberry fruits are abundant in compounds such as ellagic acid, superoxide, polysaccharides, proanthocyanidins, and superoxide dismutase, among others. Raspberries have been shown to effectively lower blood lipid levels and reduce blood glucose concentrations [11]. Proanthocyanidins (PROs), which are extracted from raspberry fruits, have gained widespread application in the treatment of diabetes. PROs exhibit properties that lower blood lipid levels and blood glucose concentrations, as well as possessing antioxidant, anti-inflammatory, anti-tumor, and antibacterial effects [12,13].

Given that the pathogenesis of DMED is believed to be driven by chronic hyperglycemia, the implementation of effective glycemic control emerges as a paramount strategy for the management of DMED [14,15]. Lipid metabolism disorders frequently co-occur with type 2 diabetes mellitus (T2DM) [16]. Research indicates that diabetes adversely affects lipid metabolism, leading to increased total cholesterol (TC) and triglyceride (TG) [16]. Hyperlipidemia is recognized as a predominant risk factor for ED [17]. Elevated blood lipid levels may increase oxidative stress, which may trigger ED [18]. Therefore, the investigation of effective pharmacological agents aimed at modulating glycolipid metabolism represents a critical avenue for managing DMED.

The phosphatidylinositol 3-kinase (PI3K)/protein kinase B (Akt) signaling pathway has been recognized as a potential target for DMED [19]. PI3K is the upstream signaling initiator within the PI3K/Akt signaling pathway. Under stress conditions, phosphatidylinositol triphosphate (PIP3) present on the plasma membrane promotes Akt complete phosphorylation. On one hand, phosphorylated Akt can directly mediate an anti-apoptotic response [20,21,22]; conversely, it can also activate its downstream effector, caspase-3, which further amplifies the anti-apoptotic capacity [23]. The penis perceives sexual stimulation and produces parasympathetic nitric oxide (nNO) and increases intracellular cyclic guanosine monophosphate (cGMP), which promotes erection [24,25]. Therefore, the decrease in NO synthesis and activity loss lead to DMED. Many research efforts have shown a strong connection between the production of NO and the PI3K/Akt signaling pathway [26,27].

Consequently, the primary objective of this study was to ascertain the efficacy of PROs in the treatment of DMED through the modulation of glucolipid metabolism and the PI3K/Akt signaling pathway.

## 2. Results

### 2.1. Effects of PROs on Glucose and Lipid Metabolism in DMED Rats

#### 2.1.1. Body Weight Changes

Weight loss is a prevalent characteristic associated with DM [28]. The results indicated that the body weight of rats in the DMED group was significantly lower than that of the control group (Figure 1A). In contrast, the body weight of rats in the PRO-treated group was significantly elevated relative to the DMED group. The body weight of rats in the tadalafil (TAD) group did not exhibit any significant difference compared to that of the DMED group.

#### 2.1.2. Blood Glucose Levels

After treatment, the blood glucose levels of rats in the DMED group were significantly elevated compared to the control group (Figure 1B). In comparison to the DMED group, PROs treatment decreased the blood glucose levels, whereas TAD did not exhibit any significant hypoglycemic effect (Figure 1B). The results demonstrated that PROs exhibited a notable hypoglycemic effect.

#### 2.1.3. Effects of PROs on Serum Insulin and Liver Glycogen Levels in DMED Rats

Insufficient serum insulin secretion constitutes a significant pathogenic factor in the development of diabetes mellitus [29]. The liver serves as the central hub for glucose and lipid metabolism. Liver glycogen functions as the primary source of blood glucose [30,31]. Furthermore, the equilibrium between hepatic glycogen synthesis and catabolism is a crucial determinant in sustaining stable blood glucose levels [30,31]. The results indicated that serum insulin and liver glycogen levels were lower in the group of rats used as a model than in the control group (Figure 1C,D). In comparison to the model group, there was a notable enhancement in serum insulin and liver glycogen levels in rats that received treatment with PROs (Figure 1C,D).

#### 2.1.4. Effects of PROs on Blood Lipid Levels in DMED Rats

As shown in Figure 1E, the TC, TG, and low-density lipoprotein (LDL) levels of rats in the model group were elevated and high-density lipoprotein (HDL) levels were decreased compared with the control group. After treatment, PROs decreased TC, TG, and LDL levels and increased HDL levels. TAD had no effect on blood lipids in DMED rats.

Based on these findings, it was observed that PROs exerted a superior regulatory effect on body weight, blood glucose levels, serum insulin, hepatic glycogen, and blood lipids in DMED rats. This suggests that PROs significantly improve glycolipid metabolism in DMED rats.

### 2.2. Erectile Function Assessment

The ratio of maximum intracavernosal pressure (ICP)/mean arterial pressure (MAP) is a reliable indicator for the assessment of erectile function [32]. Our results showed that the ICP and ICP/MAP ratios in rats of the DMED group were significantly lower than those observed in the control group (Figure 2A,C). Both ICP and ICP/MAP ratios exhibited significant improvements in rats subjected to PROs and TAD treatment compared to their untreated counterparts (Figure 2A,C).

### 2.3. Effects of PROs on Sperm Quality in DMED Rats

The adverse effects of diabetes on sperm quality have been extensively documented [33]. Sperm concentration and survival rate are critical parameters of fertility function in rats. Sperm concentration and survival rates were significantly reduced in DMED rats compared to those in the control group (Figure 3A,B). Treatment with PROs and TAD resulted in significant improvements in sperm concentration and survival rates in rats (Figure 3A,B).

### 2.4. Effects of PROs on Endothelial Function in DMED Rats

Penile erection is a neuroendocrine-regulated vasopressor response characterized by diastolic changes, which can be modulated by vascular endothelial function [34]. NO, endothelin-1 (ET-1), and lectin-like oxidized low-density lipoprotein receptor-1 (LOX-1) are critical biomarkers of vascular endothelial function. The results of the enzyme-linked immunosorbent assay (ELISA) revealed a significant decrease in serum NO levels (Figure 4A), accompanied by a notable increase in ET-1 and LOX-1 levels within the model group (Figure 4B,C). In comparison to the model group, levels of NO were significantly elevated (Figure 4A), while levels of ET-1 and LOX-1 were significantly reduced in the PRO- and TAD-treated rats (Figure 4B,C).

### 2.5. Effects of PROs on Sex Hormones in DMED Rats

Endothelial function in males is modulated by sex hormones [35]. As illustrated in Figure 5, levels of testosterone (T) in the model group were significantly lower than those in the control group. T levels in the PROs group were markedly elevated compared to the model group. Nevertheless, the analysis revealed no statistically significant differences in the levels of luteinizing hormone (LH) and follicle-stimulating hormone (FSH) between the groups.

### 2.6. Testis Histopathological Changes

As shown in Figure 6, the testis in the control group exhibited no discernible morphological abnormalities, characterized by tightly organized seminiferous tubules and spermatogonia, as well as the presence of spermatozoa within the lumen. In the testis of DMED rats, the spermatogonia within the seminiferous tubules were dislodged, obstructing the lumen and exhibiting an irregular arrangement. Additionally, there was necrosis of spermatocytes, fragmentation of cytoplasmic nuclei, and a significant reduction in the number of spermatozoa. Conversely, when compared to the model group, the testicular tissues of the PROs and TAD treatment groups tended normalization, including the restoration of the seminiferous tubule, spermatogenic cell structures, and the spermatozoa count significantly increased, and high doses of PROs markedly mitigated the DM-induced testicular damage in the rats.

### 2.7. Network Pharmacology Reveals Potential Targets and Pathways for PROs Treatment of DMED

To identify potential targets for PROs in the treatment of DMED, we conducted a query for targets associated with PROs using the Pharmmapper database, as well as targets related to DMED derived from the GeneCards, OMIM, and TTD databases. In total, 162 potential drug targets and 1827 potential disease targets were identified from the databases above (Figure 7A). Subsequently, 123 common targets associated with PROs and DMED were identified through the application of a Venn diagram (Figure 7A), and a protein–protein interaction (PPI) network was constructed using the STRING database. Finally, Cytoscape software identified five key targets (AKT1, ALB, CASP3, ESR1, EGFR) of PROs that interact with DMED within the PPI network (Figure 7B).

The Gene Ontology (GO) functional annotation of the 123 common potential targets, conducted using the DAVID database, revealed 344 biological processes, 50 cellular components, and 50 molecular functions (*p* < 0.05). The GO functional annotation indicates that the biological processes are predominantly linked to signal transduction, phosphorylation, and the negative regulation of the apoptotic process (Figure 7C). The cellular components were chiefly associated with the cytosol, nucleus, and plasma membrane (Figure 7C). Furthermore, the molecular functions predominantly encompassed protein binding, identical protein binding, and ATP binding (Figure 7C). These findings imply that PROs have the potential to exert therapeutic effects through the modulation of various biological processes related to DMED.

Based on the KEGG pathway enrichment analysis, a total of 130 signaling pathways were identified (*p* < 0.05). As illustrated in Figure 7D, the PI3K-Akt signaling pathway demonstrated a significant association with the core targets. The top 30 KEGG pathways exhibiting significant enrichment were utilized to generate bubble plots.

### 2.8. Molecular Docking Results for PROs and Potential Targets

The five core targets (AKT1, ALB, CASP3, ESR1, and EGFR) were subjected to molecular docking with PROs. The results demonstrated that the binding energies were all less than −7.0 kcal/mol [36] (Figure 8A–F), indicating that PROs exhibit strong binding activity with AKT1, ALB, CASP3, ESR1, and EGFR.

### 2.9. Effects of PROs on Core Target Levels and the PI3K-Akt Pathway in DMED Rats

ELISA was used to validate the relationship between PROs and core targets (AKT1, ALB, CASP3, ESR1, and EGFR). The ELISA results demonstrated that the level of AKT1 in the penile corpus cavernosum of rats in the model group was significantly decreased, whereas the level of CASP3 exhibited a notable increase (Figure 9A). Furthermore, following PROs treatment, the level of AKT1 was elevated, while the level of CASP3 was reduced (Figure 9A). Nevertheless, no statistically significant differences were observed in the expression levels of ALB, EGFR, and ESR1 among the groups. To further corroborate the findings obtained from the ELISA assay, we subsequently conducted real-time PCR experiments. The results revealed a notable decrease in AKT levels within the penile corpus cavernosum of DMED rats, alongside a significant elevation in CASP3 levels; furthermore, PROs treatment effectively reversed the expression levels of both AKT and CASP3 (Figure 9B). Phospho-Akt (Ser473) levels in penile tissue were quantified using ELISA. The results indicated a significant reduction in pAkt levels within the model group, suggesting that Akt phosphorylation was inhibited (Figure 9C). However, subsequent to PROs treatment, pAKT levels were markedly elevated (Figure 9C). Consequently, we established that PROs have the potential to alleviate DMED by modulating the expression levels of AKT and CASP3, while concurrently influencing the PI3K-Akt signaling pathway.

## 3. Discussion

Prior investigations have elucidated that diabetes models induced by STZ injections are capable of inducing erectile dysfunction in male rats [37,38]. Consequently, in the present study, we developed a DMED rat model utilizing STZ and examined the therapeutic effects of PROs on DMED. The findings demonstrated that PROs exhibited an ameliorative impact on erectile function in DMED rats, with key mechanisms potentially associated with the modulation of glycolipid metabolism and the PI3K-Akt signaling pathway.

Blood glucose levels exert a significant influence on the severity of DMED, and the regulation of these levels constitutes a fundamental principle in the therapeutic management of DMED [39]. Hyperglycemia has the potential to induce an elevation in the levels of reactive oxygen species (ROS) [40,41], which disrupts the bioavailability of exhaled nitric oxide (eNO), leading to endothelial dysfunction. These alterations culminate in impaired vasodilation through a diverse array of mechanisms, ultimately contributing to the manifestation of erectile abnormalities [42]. The liver and pancreas are integral components in the regulation of glucose homeostasis. Upon an increase in blood glucose levels, beta cells within the pancreas release insulin. Insulin facilitates the uptake of glucose by muscle and adipose tissues while simultaneously stimulating the liver to convert excess glucose into glycogen for storage. This physiological process serves to reduce blood glucose levels and mitigate the risk of long-term hyperglycemia [43,44]. According to previous studies, impaired insulin function increases blood glucose levels [45]. The liver serves as the central hub for glucose–lipid metabolism, with liver glycogen acting as a direct source of blood glucose, which is crucial for maintaining blood glucose levels [46]. Persistent hyperglycemia may result in hepatic dysfunction, subsequently leading to metabolic disorders that diminish liver glycogen levels and consequently elevate blood glucose levels [47]. Additionally, liver glycogen levels may serve as an indicator of insulin concentrations [48]. Insulin not only facilitates glucose uptake in peripheral tissues but also reduces blood glucose levels by inhibiting gluconeogenesis in the liver. When insulin levels are elevated, glycogen synthesis in the liver is augmented, while glycogenolysis and gluconeogenesis are concurrently inhibited, thereby effectively reducing blood glucose concentrations [43,44]. Our results show that PROs significantly enhanced insulin and hepatic glycogen levels in DMED rats, thereby playing a crucial role in the maintenance of glucose homeostasis.

DM frequently presents with abnormalities in blood lipids [49]. When blood glucose metabolism is dysregulated, it results in decreased insulin utilization, thereby enhancing the activity of lipolytic hormones, which ultimately culminates in elevated blood lipid levels [50]. Adipose tissue plays a pivotal role in glucose homeostasis and significantly influences glucose metabolism primarily through the secretion of various hormones and the release of free fatty acids. Free fatty acids and glycerol released from adipose tissue are readily taken up by the liver and utilized for energy production. The availability of these substances can modulate hepatic glucose processing, thereby contributing to the enhancement of glucose homeostasis to a certain degree [51]. Dysregulations in lipid metabolism may impair the functionality of adipose tissue in secreting factors detrimental to pancreatic islet cells, consequently reducing pancreatic islet function, diminishing pancreatic secretion, and resulting in decreased insulin levels, ultimately contributing to elevated blood glucose levels [52]. The present study demonstrated a significant improvement in blood glucose and lipid levels in the PRO-treated group of rats. These findings suggest that PROs may play a role in maintaining glucose homeostasis through the regulation of blood lipids.

Dyslipidemia is significantly correlated with ED [53]. Hyperlipidemia may contribute to erectile abnormalities, with the impairment of vascular endothelial function identified as a primary factor. Hypertriglyceridemia can result in the excessive accumulation of free fatty acids, which compromises the endothelial cell integrity. Additionally, it diminishes the NO production and subsequent endothelial dysfunction. Elevated total cholesterol levels are associated with increased ROS production, thereby exacerbating oxidative stress [54]. Concurrently, hyperlipidemia contributes to the development of atherosclerosis in the small penile arteries, resulting in compromised blood circulation and diminished penile congestion, ultimately culminating in inadequate penile erection. Atherosclerotic arterial walls also produce substantial quantities of oxidized LDL and superoxide, which contribute to the inactivation of NO [55]. Hypercholesterolemia additionally induces neuronal axonal degeneration, resulting in a significant reduction in penile neuronal NO synthase (nNOS) neurons, which in turn diminishes NO utilization and adversely affects erectile function [55].

Our findings indicate that blood glucose, serum insulin, liver glycogen, TC, TG, and HDL levels in DMED rats treated with PROs were significantly elevated compared to the model group, whereas LDL levels were significantly reduced, implying that PROs enhance the regulation of glucose and lipid metabolism. ICP and ICP/MAP are critical indicators for assessing penile erectile function. The results demonstrate that both ICP and ICP/MAP in DMED rats showed significant improvement relative to the model group, indicating that PROs effectively ameliorate erectile dysfunction induced by diabetes mellitus. In conclusion, PROs have the potential to enhance erectile function in DMED by modulating glycolipid metabolism.

Penile erection is intricately regulated by endothelial function. NO is predominantly released by non-cholinergic nerves and endothelial cells, which can promote normal angiogenesis and sustain blood circulation. In patients suffering from ED, NO expression levels are notably diminished [56,57]. ET-1 is a potent vasoconstrictor. It promotes vasoconstriction and mediates the contraction of both urethral smooth muscle and cavernous smooth muscle, thereby affecting erectile function [58]. Enhanced expression of LOX-1 results in impaired endothelial vasodilation in the context of diabetes [59]. The results of this study demonstrate that PROs significantly reduce the serum expression levels of ET-1 and LOX-1, while markedly increasing the expression level of NO in DMED rats. These outcomes indicate that PROs may ameliorate vascular endothelial damage and restore vasodilatory function, thereby facilitating the recovery of erectile dysfunction in this animal model.

T deficiency is a significant etiological factor in DMED. T is primarily secreted by testicular mesenchymal stromal cells and may facilitate penile erection by modulating central pathways, activating nitric oxide synthase (NOS), and reducing cavernous vasoconstriction [60]. Endothelial function is influenced by T levels, which are believed to promote vascular endothelial function through both direct and indirect mechanisms. T enhances penile endothelial cell function and regulates endothelium-dependent vasodilation [61]. Sex hormone-binding globulin (SHBG) serves as the primary transporter of T within the bloodstream. A significant contributor to the reduction of T levels in individuals with diabetes is attributed to reduced sex hormone binding globulin [62]. The findings of this experiment indicated that the T levels in the model group were significantly diminished. Following the PROs intervention, the T levels showed a marked increase. Nevertheless, LH and FSH levels were not statistically different between groups. We examined the potential explanations for this phenomenon: LH and FSH are produced by the pituitary gland; therefore, it is plausible that PROs may improve the T-producing milieu, thereby elevating T levels. However, PROs exert no significant influence on the secretion of FSH and LH by the pituitary gland.

The testis is the site of T secretion and sperm production. The morphological integrity of the testis, along with T levels, sperm concentration, and survival rates, constitutes critical parameters that reflect male reproductive function. The DMED rats showed a reduction in sperm concentration and survival rates. Light microscopy examinations revealed detachment of testicular spermatogonial cells, necrosis of these cells, and a notable decrease in sperm counts within the model group. These findings suggest that diabetes induces significant reproductive damage in male rats. In comparison to the model group, PROs ameliorated the histopathological alterations in the testis caused by diabetes mellitus to a considerable extent, while concurrently enhancing sperm concentration and survival rates.

Key targets, including AKT1, ALB, CASP3, ESR1, and EGFR, were identified through network pharmacology analysis. Both AKT1 and CASP3 were found to be significantly enriched within the PI3K-Akt signaling pathway. The pathological process of DMED is involved and regulated by the PI3K-Akt signaling pathway [63]. Akt1 is central to facilitating cell division, promoting angiogenesis, and preventing apoptosis [64]. Furthermore, Akt1 can enhance NO synthesis, which is crucial for the regulation of penile erection [65]. CASP3 is recognized as a hallmark protein associated with the process of apoptosis. As DM progresses, disruptions in glucose and lipid metabolism precipitate atherosclerosis in significant arteries and induce both vascular and microvascular pathologies. These pathological changes culminate in ischemia and hypoxia within the cavernous body, resulting in reduced smooth muscle cells, an increase in fibrosis, and a heightened rate of apoptosis in the cavernous body. Collectively, these factors significantly contribute to the onset of ED. Numerous studies have substantiated the critical role of apoptosis in the pathogenesis of DMED [66,67]. The activation of the PI3K-Akt signaling pathway has been shown to attenuate the pro-apoptotic effects of CASP3 through the modulation of its phosphorylation status and subsequent inactivation of this apoptosis-associated factor [63]. Molecular docking analyses demonstrated that PROs exhibit a robust affinity for both AKT1 and CASP3. Subsequent ELISA assays corroborated these findings, revealing an elevation in AKT1 levels alongside a reduction in CASP3 levels within the penises of rats treated with PRO. Similarly, AKT was downregulated, and CASP3 upregulated in the penises of DMED rats, according to real-time PCR results. The results obtained from the ELISA demonstrated a significant inhibition of AKT phosphorylation. However, treatment with PROs effectively reversed this phenomenon. Nevertheless, the expression levels of ALB, ESR1, and EGFR did not differ significantly across all rat groups, despite their identification as central targets in the network pharmacological analysis. This phenomenon can be elucidated by two primary factors: firstly, the targets identified through cyberpharmacology, while grounded in databases and computational models, do not guarantee biological relevance in every instance, thereby allowing for the possibility of false positives; secondly, the intricate nature of signaling pathways within organisms implies that interactions among multiple targets and pathways may obscure the effects attributable to a singular target. Such findings warrant further investigation in subsequent research endeavors. In conclusion, our findings indicate that PROs may modulate the PI3K-Akt signaling pathway by upregulating AKT levels and downregulating CASP3 levels in rat penile tissue, which may have therapeutic effects on DMED.

## 4. Materials and Methods

### 4.1. Animals

A total of one hundred 6-week-old male Sprague-Dawley rats, with an average body mass of 200 ± 20 g, were procured from the Department of Experimental Animals at Xi’an Jiaotong University (License Number: SCXK [Shaanxi] 2018-001).

### 4.2. DMED Rat Model and Experimental Designs

Ten rats were randomly selected to serve as the control group. Following this, 2 g of streptozotocin (STZ) (Beijing Solabao Technology Co., Ltd., Beijing, China) was dissolved in 200 mL of citric acid–sodium citrate buffer in an ice bath to create an STZ solution (10 mg/mL), followed by an intraperitoneal injection of 60 mg/kg within 30 min [68]. After 72 h, blood was collected from the tail tip for blood glucose measurement, and a blood glucose value > 16.7 mmol/L was considered as successful DM modelling [69]. Two weeks following the successful modeling of DM rats, the assessment of erectile function was conducted. An apomorphine (APO) (Sigma-Aldrich Chemical Company, St. Louis, MO, USA) solution (10 μg/mL), consisting of 5 mg APO, 250 mg vitamin C, and 500 mL of saline, was administered via injection into the loose skin of the rats’ necks at a dosage of 100 μg/kg. Erection was observed within 30 min (glans engorged, penile body end exposed). Those without erection were considered to have DMED. The DMED rats were randomly divided into five groups (*n* = 10): the model group, PROs high-dose group (150 mg/kg/day, i.g.), PROs medium-dose group (100 mg/kg/day, i.g.), PROs low-dose group (50 mg/kg/day, i.g.), and TAD (Macklin, Shanghai, China) (0.52mg/kg/day, i.g.) treatment group. Blood glucose levels were measured, and erectile function was assessed at the end of the dosing period. Biological samples were collected for further study.

### 4.3. Glucose Determination

Blood was collected from the rat tails before and after treatment, and blood glucose was measured using a blood glucose meter (Lifescan, Johnson & Johnson, Shanghai, China).

### 4.4. Assessment of Erectile Function

ICP, MAP, and ICP/MAP were measured 4 weeks after administration. The rats were anesthetized before measurement. The right common carotid artery was carefully isolated, and a PE-50 tube containing 250 IU/mL of heparin saline (Beijing Coolaber Technology Co., Ltd., Beijing, China) was inserted, with a pressure transducer affixed to the caudal end to facilitate the determination of MAP. The perineum of the rat’s abdomen was incised medially to expose the penile corpus cavernosum, into which a 25-gauge needle attached to the PE-50 tube was inserted to measure ICP. Both were connected to a Powerlab working platform (ADInstruments Company, Sydney, Australia) (5 V voltage, 15 HZ frequency, 1.2 ms wave width, 1 min duration, 3-min-interval electrical stimulation) to observe ICP/MAP.

### 4.5. Serum Measurements

Rats were anaesthetized, blood was taken from the abdominal aorta, centrifuged (3000 r/min) for 15 min, and the supernatant was removed. Serum insulin, hepatic glycogen, TG, TC, HDL, and LDL levels were measured using a test kit (glycogen test kit, TG test kit, TC test kit, HDL test kit, and LDL test kit (Nanjing Jianjian Bioengineering Institute, Nanjing, China)). The expression levels of sex hormones, including FSH, LH, and T, and indicators of vascular endothelial function, such as NO, ET-1, and LOX-1, were quantified using ELISA. ELISA kits for FSH, LH, T, NO, ET-1, and LOX-1 were purchased from Wuhan Eliot Biotechnology Co., Ltd., Wuhan, China. The assay was performed as instructed.

### 4.6. Histopathologic Analysis

One side of the rat testis was excised, immediately fixed in 4% paraformaldehyde, subsequently embedded in paraffin, sectioned, routinely stained with HE, and histologically examined under a microscope.

### 4.7. Semen Quality Analysis

The epididymis was excised from one side of the rat and subsequently immersed in a medium containing 1 mL of preheated benchtop solution at 37 °C. Following this, the spermatozoa were liberated from the epididymis by gentle agitation using a pipette gun. Next, 10 μL of the upper layer of the resulting thick white liquid was carefully aspirated into a centrifuge tube containing 990 μL of preheated benchtop solution for dilution. Subsequently, a 10 μL diluted solution was placed onto a sperm-counting plate, and the concentration and survival rates of the sperm were determined using a computer-assisted semen analysis (CASA) system (Beijing Weili New Century Technology Development Co., Ltd., Beijing, China).

### 4.8. Network Pharmacological Analysis

#### 4.8.1. Target Prediction of PRO

The SDF (structure data file) files associated with PROs were retrieved from the PubChem database (https://pubchem.ncbi.nlm.nih.gov/, accessed on 3 August 2024) and subsequently imported into the Pharmmapper database (https://lilab-ecust.cn/pharmmapper/index.html, accessed on 3 August 2024) to facilitate the prediction of potential targets for PRO.

#### 4.8.2. Collection of Disease Targets

Gene Cards (https://www.genecards.org, accessed on 3 August 2024), OMIM (https://www.omim.org/, accessed on 3 August 2024), and TTD (https://db.idrblab.net/ttd/, accessed on 3 August 2024) databases were utilized to investigate disease targets pertaining to the keyword “Diabetic erectile dysfunction”. The intersection of the targets associated with PROs and those related to DMED was identified and subsequently represented in a Venn diagram.

#### 4.8.3. Construction of the Protein–Protein Interaction Network

The PPI network was established by importing the potential target genes into the STRING database (https://www.stringdb.org/, accessed on 3 August 2024). The species was designated as “Homo sapiens”, with the minimum interaction threshold configured to “highest confidence”, while the non-essential target genes were concealed. The PPI data were subsequently imported into Cytoscape (Cytoscape 3.9.0), where the parameters of Degree, Betweenness, and Closeness were computed to identify topologically significant nodes. The nodes exhibiting the top five target genes based on their degree were identified as core target genes.

#### 4.8.4. Analysis of GO and KEGG Pathway Enrichment

GO functional annotation and KEGG pathway analysis were conducted on potential target genes related to PROs treatment for DMED utilizing the DAVID database (https://david.ncifcrf.gov/summary.jsp, accessed on 3 August 2024).

### 4.9. Molecular Docking

The molecular structure of PROs was retrieved from the PubChem database (https://pubchem.ncbi.nlm.nih.gov/, accessed on 3 August 2024). The protein structures corresponding to the core targets were sourced from the PDB database (https://www.rcsb.org/, accessed on 3 August 2024). The molecular structure of PROs was initially pre-processed to incorporate hydrogenation and charge adjustments, while the MMFF94 force field was calibrated for energy minimization. The protein structures were pre-processed utilizing PyMOL (PyMOL 2.2.0), which involved the removal of water molecules and non-liganded small molecules. Molecular docking was conducted utilizing AutoDock Vina, and the resulting docking outcomes were subsequently analyzed and visualized using PyMOL (PyMOL 2.2.0).

### 4.10. Validation of Compounds by Experiments

The protein levels of AKT1, ALB, CASP3, ESR1, and EGFR (Wuhan Elerite Biotechnology Co., Ltd., Wuhan, China) in rat penile corpus cavernosum were assessed using ELISA kits. Subsequently, the results of ELISA were validated by real-time PCR. Primers are shown in Table 1. RNA was extracted from rat penile corpus cavernosum using the Trizol method, followed by reverse transcription. RNA was extracted from rat penile corpus cavernosum using the Trizol method, followed by reverse transcription. The resulting cDNA was amplified under specific conditions (37 °C, 15 min; 85 °C, 5 s; and 4 °C, 60 min). PCRs were performed as described in this manual using a real-time PCR program. The mRNA expression was determined by 2^−ΔΔCT^. Finally, phospho-Akt (Ser473) levels were determined using ELISA assay kits (Wuhan Elerite Biotechnology Co., Ltd., Wuhan, China).

### 4.11. Statistical Analysis

SPSS 23.0 was used for statistical analysis in this study. One-way analysis of variance (ANOVA) was used for comparison between groups (significance *p* < 0.05).

## 5. Conclusions

Our study represents the inaugural assessment of the therapeutic effects of PROs on DEMD. The evaluation of erectile function substantiated that PROs significantly enhance erectile function in DEMD rats. Our research elucidates that PROs may effectively treat DMED through diverse mechanisms: (1) PROs can modulate glucose and lipid metabolism in DMED rats through the regulation of blood glucose, blood lipid, liver glycogen, and serum insulin levels. (2) PROs can enhance vascular endothelial function in DMED rats by augmenting NO levels while simultaneously reducing the levels of ET-1 and LOX-1. (3) PROs may elevate T levels in DMED rats, ameliorate diabetes-induced pathological damage to the testis, and enhance sperm concentration and survival rates, thereby improving reproductive function in these animals. (4) Utilizing network pharmacological screening, molecular docking, ELISA, and real-time PCR validation, this study confirms that PROs may exert a therapeutic effect by enhancing AKT levels and reducing CASP3 levels, thereby regulating the PI3K-Akt signaling pathway. Our findings indicate that PROs are a viable therapeutic for DMED; however, further investigations are essential to explore their clinical implications.

## Figures and Tables

**Figure 1 ijms-25-11004-f001:**
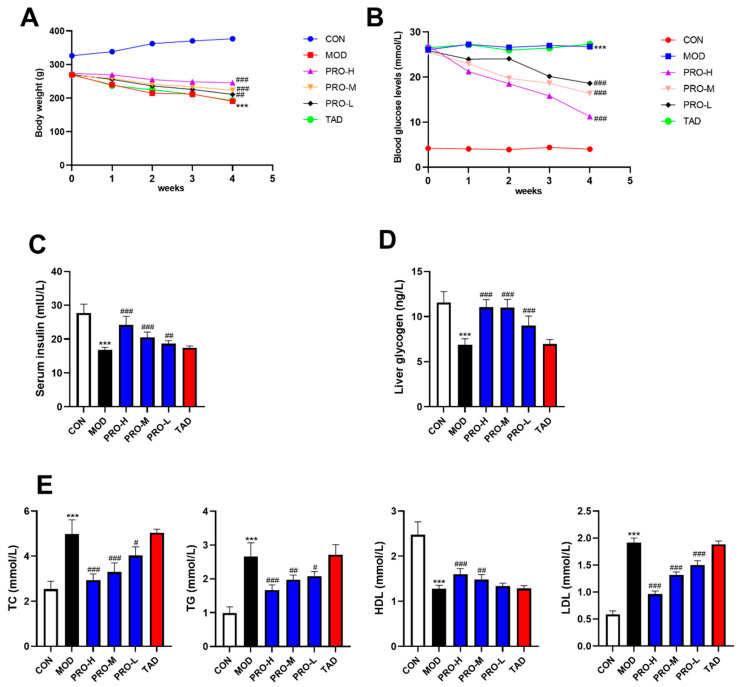
Effects of PROs on body weight, glucose, and lipid metabolism in DMED rats. (**A**) Changes in body weight of rats after treatment. (**B**) Changes in blood glucose levels of rats after treatment. (**C**) Serum insulin levels of rats in each group. (**D**) Liver glycogen levels of rats in each group. (**E**) Blood lipid (TC, TG, HDL, LDL) levels of rats in each group. Sixty rats were randomly allocated into six groups: the control group (CON), model group (MOD), PROs high-dose group (PRO-H), PROs medium-dose group (PRO-M), PROs low-dose group (PRO-L), and tadalafil (TAD) group. The data are expressed as the means ± SDs (*n* = 10). *** *p* < 0.001 vs. the CON group; ^###^ *p* < 0.001, ^##^ *p* < 0.01, ^#^
*p* < 0.05 vs. the MOD group. TC, total cholesterol; TG, triglyceride; HDL, high-density lipoprotein; LDL, low-density lipoprotein.

**Figure 2 ijms-25-11004-f002:**
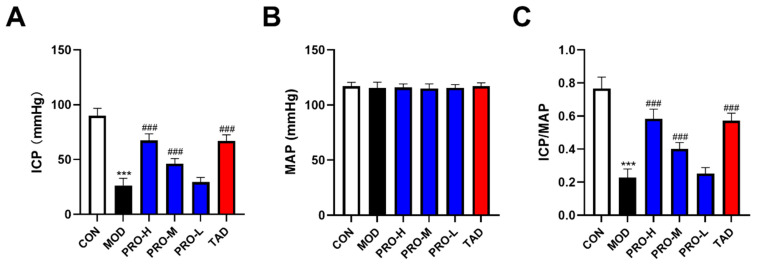
Erectile function assessment of rats in each group after treatment. ICP, MAP, and ICP/MAP were measured 4 weeks after administration. (**A**) ICP of rats in each group. (**B**) MAP of rats in each group. (**C**) ICP/MAP ratios of rats in each group. The data are expressed as the means ± SDs (*n* = 10). *** *p* < 0.001 vs. the CON group; ^###^ *p* < 0.001 vs. the MOD group. ICP, intracavernosal pressure; MAP, mean arterial pressure.

**Figure 3 ijms-25-11004-f003:**
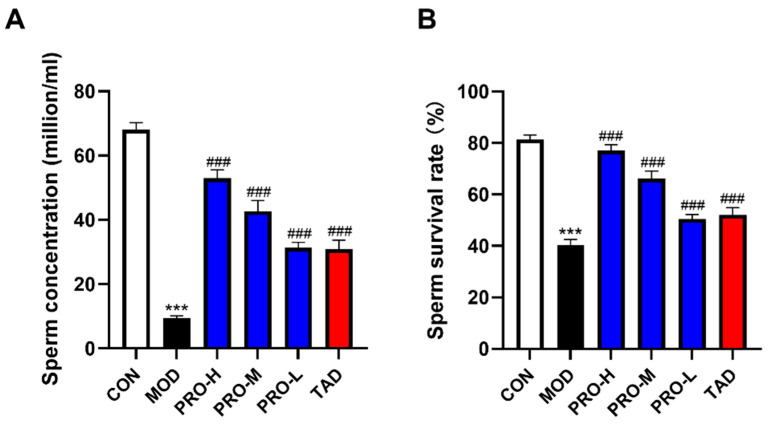
Effects of PROs on sperm quality in DMED rats. The epididymis of rats was collected, and subsequent analyses were conducted to determine sperm concentration and survival rates. (**A**) Sperm concentration of rats in each group. (**B**) Sperm survival rates of rats in each group (*n* = 10). The data are expressed as the means ± SDs. *** *p* < 0.001 vs. the CON group; ^###^ *p* < 0.001 vs. the MOD group.

**Figure 4 ijms-25-11004-f004:**
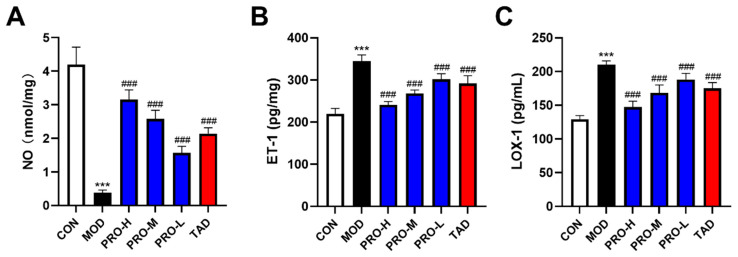
Effects of PROs on endothelial function in DMED rats. (**A**) NO levels of rats in each group. (**B**) ET-1 levels of rats in each group. (**C**) LOX-1 levels of rats in each group. The data are expressed as the means ± SDs (*n* = 10). *** *p* < 0.001 vs. the CON group; ^###^ *p* < 0.001 vs. the MOD group. ET-1, of rats in each group; LOX-1, lectin-like oxidized low-density lipoprotein receptor-1.

**Figure 5 ijms-25-11004-f005:**
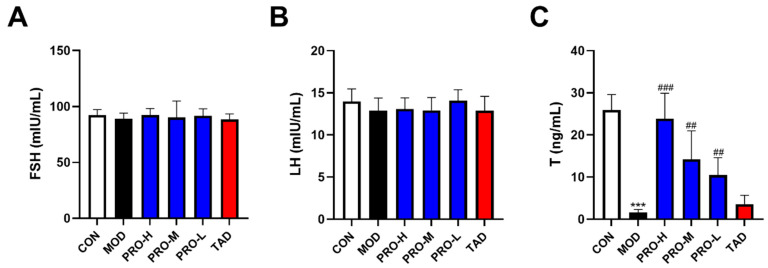
Sex hormone levels of rats in each group after treatment. (**A**) FSH levels of rats in each group. (**B**) LH levels of rats in each group. (**C**) T levels of rats in each group. The data are expressed as the means ± SDs (*n* = 10). *** *p* < 0.001 vs. the CON group; ^###^ *p* < 0.001, ^##^ *p* < 0.01 vs. the MOD group. FSH, follicle-stimulating hormone; LH, luteinizing hormone; T, testosterone.

**Figure 6 ijms-25-11004-f006:**
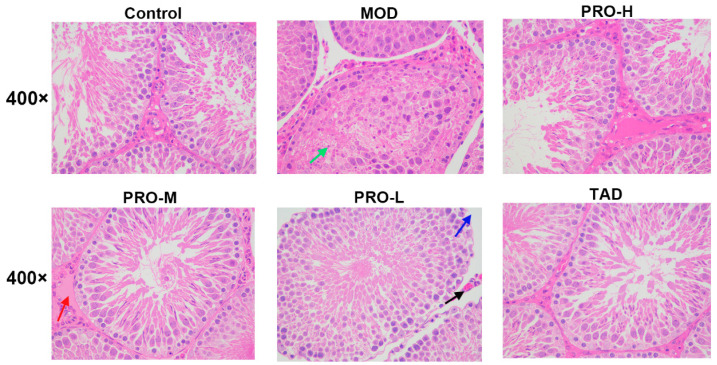
Hematoxylin and eosin (HE) staining of rat testis. Rats in the control group exhibited an intact testicular cell structure, tightly organized spermatogonia, and the presence of spermatozoa within the lumen without any apparent pathological alterations. Rats in the model group displayed dislodged spermatogonia, necrosis of spermatocytes, and a significant reduction in the number of spermatozoa, and distinct pathological changes. Conversely, the testicular tissues of the PROs and TAD treatment groups tended normalization. The green arrow indicates spermatocyte necrosis and nuclei consolidated or fragmented. The red arrow indicates eosinophilic tissue fluid exudation. The blue arrow signifies a sparsely arranged varicocele with widened gaps. The black arrow indicates interstitial capillary stasis.

**Figure 7 ijms-25-11004-f007:**
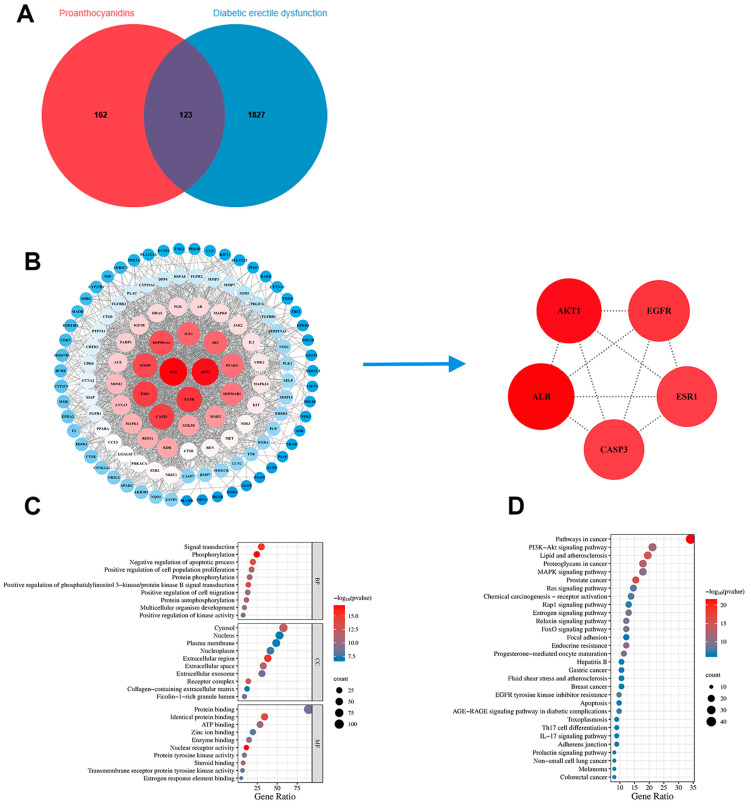
Network pharmacology reveals potential targets and pathways for PROs treatment of DMED. (**A**) Wayne analysis of PROs and DMED intersection targets. (**B**) PPI network diagram of intersection target of PROs and DMED. The size and coloration of the circles signify the extent of correlation with other targets; specifically, circles that are larger and redder indicate a stronger correlation. (**C**) GO function enrichment analysis results. (**D**) KEGG pathway enrichment analysis results.

**Figure 8 ijms-25-11004-f008:**
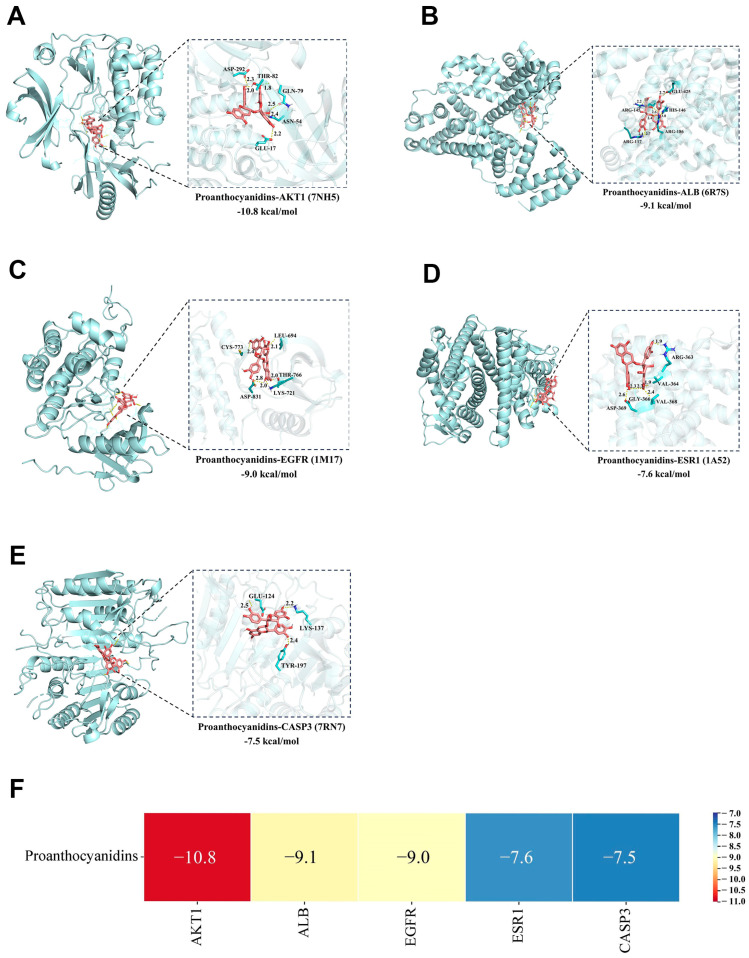
Molecular docking results. (**A**–**E**) Schematic diagram of the binding mode and details of the binding pose of PROs to AKT1, ALB, EGFR, ESR1, and CASP3. (**F**) Molecular docking results heat map of PROs and key targets.

**Figure 9 ijms-25-11004-f009:**
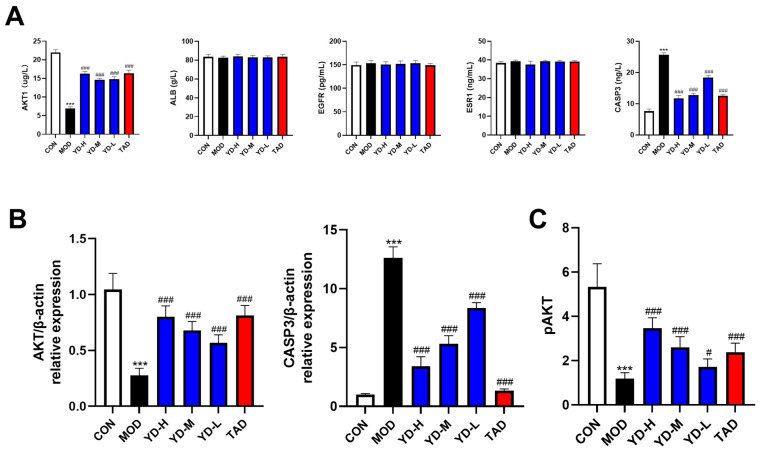
Effects of PROs on core target levels and the PI3K-Akt Pathway in DMED rats. (**A**) Evaluation of AKT1, ALB, CASP3, ESR1, and EGFR expression in the penile corpus cavernosum of rats (*n* = 10). (**B**) Relative mRNA expression levels of AKT and CASP3 (*n* = 5). (**C**) pAKT levels in the penile corpus cavernosum of rats (*n* = 10). The data are expressed as the means ± SDs. *** *p* < 0.001 vs. the CON group; ^###^ *p* < 0.001, ^#^ *p* < 0.05, vs. the MOD group.

**Table 1 ijms-25-11004-t001:** PCR primer information.

Primer	Forward Primer	Reverse Primer
AKT	GTGCTGGAGGACAATGACTACGG	AGCAGCCCTGAAAGCAAGGA
CASP3	ACTGGAAAGCCGAAACTCTTCATCA	GGAAGTCGGCCTCCACTGGTATC
*β*-actin	GGCATCCTGACCCTGAAGTA	AGGAAGGAAGGCTGGAAGAG

## Data Availability

All the datasets in this study can be provided upon reasonable request.

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
