# Peer review of "Therapeutic Effects of Proanthocyanidins on Diabetic Erectile Dysfunction in Rats"

_ijms, 2024, doi:10.3390/ijms252011004_

Round 1

Reviewer 1 Report

Comments and Suggestions for Authors

In this study, Zeng et al. investigated the effect of PRO on diabetic erectile dysfunction (DMEM) in rats. The found that PRO improves glucose and lipid metabolism as well as vascular endothelial function in DMEM rats. They proposed that the beneficial effect of PRO may be mediated by PI3K-Akt signaling pathways. Although the effect of PRO on DMEM is solid, the mechanistic study is very weak. I have following suggestions and questions:

1.     PRO has a beneficial effect on STZ-induced glucose homeostasis and insulin secretion. The authors should discuss the mechanisms by which PRO improves glucose homeostasis such as in livers, adipose tissues, or pancreas.

2.     The authors include the TAD in their study. TAD improves DMED but not glucose homeostasis, whereas PRO improves both DMED and glucose homeostasis. Whether the underlying mechanisms of TAD and PRO-mediated DMED improvement are similar or different? How much does PRO-regulated glucose homeostasis contribute to DMEM improvement?

3.     The authors mentioned that apoptosis marker CASP3 is a target of PRO. They detected the mRNA expression levels of CASP3. How about the protein levels of CASP3, especially cleaved CASP3?

4.     It seems that the endothelial function is a key in DMED. Whether PRO or TAD increases the secretion of NO in endothelial cells through PI3K-Akt signaling, thereby increasing serum NO levels?

5.     The authors only detected the AKT mRNA levels. AKT phosphorylation at S473 or T308 indicates the activity of PI3K-Akt signaling. How about the phosphorylation levels of AKT in the experiment group of Fig. 9B

6.     The authors should indicate the n number in the legend.

Author Response

Thank you very much for taking the time to review my manuscript. Answers to all questions are labeled green in the revised manuscript. We hope that the responses to your comments are compliant and clearly explained. Please see the attachment.

Reviewer 2 Report

Comments and Suggestions for Authors

The presented manuscript shows an investigation of the effect of Proanthocyanidins on a murine model of diabetes-related erectile dysfunction. The results presented show beneficial effects on the characteristics of diabetes and on erectile dysfunction; then the authors perform a network pharmacology analysis to try to explain the mechanism of action of Proanthocyanidins in this disease. Finally, the authors highlight the beneficial effects of this molecule. 

However, there are some points to consider in the research.

1. Check that the references are adequate, for example reference 5.

2. Enlarge Figure 7 to make it easier to read.

3. It is not clear to me what is the origin of the samples used for the ELISA “rat penile corpus cavernosum” the sample was tissue? explain. 

4. The authors should discuss why in the ELISA test they did not find a statistical difference for ALB, EGFR, and ESR1; although they were previously identified as hub genes in the network pharmacology analysis.

5. Similarly, the authors should discuss the lack of significant difference for FSH and LH hormones. 

Author Response

Thank you very much for taking the time to review my manuscript. Answers to all questions are labeled red in the revised manuscript. We hope that the responses to your comments are compliant and clearly explained. Please see the attachment.

Round 2

Reviewer 1 Report

Comments and Suggestions for Authors

The authors address my questions.

Author Response

Thanks a lot for the reviewer’s appreciation.

Reviewer 2 Report

Comments and Suggestions for Authors

The authors have satisfactorily answered most of my observations. However, reference 5 and those they recently added (6-8) are reviews or in vitro studies, but are not surveys, as mentioned by the authors in the text. 

Author Response

Thank you very much for taking the time to review my manuscript. The main issues we've answered that in part 2 below. Answers to all questions are labeled red in the revised manuscript. We hope that the responses to your comments are compliant and clearly explained.

Point-by-point response to Comments and Suggestions for Authors.

Comments and Suggestions for Authors

The authors have satisfactorily answered most of my observations. However, reference 5 and those they recently added (6-8) are reviews or in vitro studies, but are not surveys, as mentioned by the authors in the text. 

 Response : Thanks to your suggestion, we've updated the references. (lines 40, 550-562, pages 1, 16-17 ).

  1. Demanou,M.C.D.;Njonnou, S.R.S.; Fouda, A.A.B.; Balti, E.; Lekpa, F.K.; Ouankou, CN.; et al. Frequency and determinants of phytotherapy use in patients with type 2 diabetes in the Dschang Health District, Cameroon: a cross-sectional study. Pan Afr Med J. 2024, 47, 174. http://dx.doi.org/10.11604/pamj.2024.47.174.41677.
  2. Ghorat,F.;Mosavat, S.H.; Hadigheh, S.; Kouhpayeh, S.A.; Naghizadeh, M.M.; Rashidi, A.A.; et al. Prevalence of Complementary and Alternative Medicine Use and Its Associated Factors among Iranian Diabetic Patients: A Cross-Sectional Study. Curr Ther Res Clin Exp. 2024, 100, 100746. http://dx.doi.org/10.1016/j.curtheres.2024.100746.
  3. Ilhan,M.;Demir, B.; Yüksel, S.; Çataklı, S.A.; Yıldız, R.S.; Karaman, O.; et al. The use of complementary medicine in patients with diabetes. North Clin Istanb. 2016, 3, 34-8. http://dx.doi.org/10.14744/nci.2016.63825.
  4. Jafari,A.;Movahedzadeh, D.; Barsalani, F.R.; Tehrani, H. Investigation of attitude, awareness, belief, and practice of complementary and alternative medicine among type 2 diabetic patients: a cross sectional study. J Diabetes Metab Disord. 2021, 20, 477-84. http://dx.doi.org/10.1007/s40200-021-00769-4.
  5. Kifle,Z.D. Prevalence and correlates of complementary and alternative medicine use among diabetic patients in a resource-limited setting. Metabol Open. 2021, 10, 100095. http://dx.doi.org/10.1016/j.metop.2021.100095.